Two new species of Notodasus Fauchald, 1972 (Annelida: Capitellidae) from the Central Indo-Pacific region

Lin Junhui 1 linjunhui@tio.org.cn
García-Garza María Elena 2
http://orcid.org/0000-0001-6961-8771 Arbi Ucu Yanu 3
Wang Jianjun 1 wangjianjun220@tio.org.cn
1 Third Institute of Oceanography, Ministry of Natural Resources , Xiamen, Fujian , China
2 Facultad de Ciencias Biológicas, Universidad Autónoma de Nuevo León , Nuevo León , México
3 Research Center for Oceanography, Indonesian Institute of Sciences , Jakarta , Indonesia
Gandini Patricia
Electronic publication date: 2019 Sep 6
Publication date: 2019
Volume: 7
Electronic Location ID: e7638
Received 2019 May 28; Accepted 2019 Aug 6
Copyright: © 2019 Lin et al.
Copyright year: 2019
Copyright holder: Lin et al.
License: This is an open access article distributed under the terms of the Creative Commons Attribution License, which permits unrestricted use, distribution, reproduction and adaptation in any medium and for any purpose provided that it is properly attributed. For attribution, the original author(s), title, publication source (PeerJ) and either DOI or URL of the article must be cited.
License URL: https://creativecommons.org/licenses/by/4.0/

Keywords: Polychaeta, Taxonomy, Sulawesi Island, Notodasus, Southern China, The Central Indo-Pacific

Funding: China-Indonesia Maritime Cooperation Fund project “China-Indonesia Ecological Station Establishment and Marine Biodiversity Survey in North Sulawesi Sea” and the Public Science and Technology Research Funds Projects of Ocean 201505004 This work was supported by the China-Indonesia Maritime Cooperation Fund project “China-Indonesia Ecological Station Establishment and Marine Biodiversity Survey in North Sulawesi Sea” and the Public Science and Technology Research Funds Projects of Ocean (No. 201505004). The funders had no role in study design, data collection and analysis, decision to publish, or preparation of the manuscript.

==============================
Notodasus Fauchald, 1972 is a small genus of the polychaete family Capitellidae, including 10 described species worldwide. The genus is unusual in the Central Indo-Pacific, and there is no taxonomic record of Notodasus in this region. In this study, two new species of Notodasus are described and illustrated, namely Notodasus celebensis sp. nov. and N. chinensis sp. nov. The former species, collected from the mixed-species seagrass beds in the Indonesian island of Sulawesi, is mainly characterized by the longitudinally striated epithelium on thoracic segments and the completely separated notopodial lobes. The latter species, obtained from coastal waters off southern China, differs from its congeners with the following characters: tessellated epithelium present on anterior thorax as well as on the dorsum of chaetigers 11 and 12, notopodial lobes fused and chaetal fascicles almost touching each other on anterior abdomen, and branchial pores evident from anterior abdomen. Comparisons are made with closely related species in this paper, and a revised key is provided to all described Notodasus species. The descriptions of the two new species represent the first record of Notodasus in this region and expand the geographical distribution of the genus.

Introduction

Polychaetes, known as an ecologically important taxon of benthic macrofauna, are frequently found and numerically dominated in marine surveys. They exhibit high taxonomic diversity with over 11,000 valid species worldwide (Pamungkas et al., 2019). Of all known polychaete families, Capitellidae Grube, 1862 is a family with approximately 200 described species, represented by 43 genera (Magalhães & Blake, 2017; García-Garza, De León-González & Tovar-Hernández, 2019). This family is usually associated with organically enriched and disturbed sediments (Magalhães & Blake, 2017), and as such, some species can be used as environmental indicators (Pearson & Rosenberg, 1978; Reish, 1980; Warren, 1991). Although extensive taxonomic studies on capitellid polychaetes have been carried out, correct identification of capitellid species is fairly challenging due to their simple morphology and the change in chaetal arrangement during ontogeny (Blake, 2000; Hutchings, 2000). With the advancements in high-resolution microscopes and molecular techniques, the taxonomy of capitellid polychaetes has been extensively improved, and additional 44 new species have been added to the family since 2000, based on statistics from WoRMS (Read & Fauchald, 2018).

The taxonomic study varies in different capitellid genera, and until now, the study of several genera is scarce and confined to limited localities. For example, the majority of Notodasus species (six out of 10 species) were described from the North American coasts. Notodasus was initially established by Fauchald (1972) for the type species Notodasus magnus from the Gulf of California, mainly characterized by the presence of only capillaries on all 11 thoracic chaetigers as well as on first two abdominal chaetigers. Recently, García-Garza, Hernández-Valdez & De León-González (2009) reviewed the genus by re-examining type materials from different museums and described four additional species. In this review, the authors also proposed more diagnostic characters to differentiate species within the genus, including the epithelial texture of thorax, the degree of fusion of notopodial lobes on anterior abdomen, the shape of hooded hooks, size of abdominal lateral organs, and the methyl green staining pattern (MGSP) (Magalhães & Blake, 2017). Notodasus closely resembles Dodecaseta, the latter genus erected by McCammon & Stull (1978) and its generic definition expanded by Green (2002). These two genera overlap in generic diagnosis, and the minor morphological differences are that the former has the first two abdominal chaetigers with only capillaries while the latter bears the first one or two abdominal chaetigers with only capillaries. Recently, García-Garza, De León-González & Harris (2017) regarded Dodecaseta as a junior synonym of Notodasus, due to the high morphological similarity between these two genera. To date, 10 valid species are known in the genus (Fig. 1), and they are described from several localities: six species recorded from the North American coasts, namely N. dexterae Fauchald, 1973, N. harrisae García-Garza et al., 2009, N. hartmanae García-Garza et al., 2009, N. magnus Fauchald, 1972, N. oraria (McCammon & Stull, 1978), and N. salazari García-Garza et al., 2009; two species found in the Andaman Sea, Thailand, namely N. eibyejacobseni (Green, 2002) and N. fauchaldi (Green, 2002); N. arenicola Hartmann-Schröder, 1992 and N. dasybranchoides Magalhães & Bailey-Brock, 2012 described from Ascension Island in the central Atlantic Ocean and the Hawaiian Islands, respectively.

Figure 1 Type localities of described species of Notodasus worldwide.

Currently, a complete knowledge on the overall species diversity within the genus and its distributional range worldwide is still lacking, since the records of Notodasus species were only limited to few localities. In the Central Indo-Pacific, Notodasus is an unusual capitellid genus, and least studied until now, as evidenced by a lack of taxonomic record of its occurrence in this region. In this study, specimens of the genus were collected from the southern coasts of China (Fig. 2A) and from mixed-species seagrass beds in the Indonesian island of Sulawesi (Fig. 2B), respectively, representing the report of Notodasus in Indonesian and Chinese waters for the first time. During the taxonomic study of the material, two new species are described and illustrated herein. Detailed comparisons are made with closely related species. This study serves as a new contribution to unveil the hidden diversity of the genus Notodasus in the Central Indo-Pacific. The description of the two new species also allows us to better understand the geographical distribution of the genus. A revised key to all Notodasus species is also provided in this paper.

Figure 2 Map of survey areas.

(A) sampling stations in the Qinzhou Bay, the southern coast of China, and (B) sampling stations in mixed-species seagrass beds of northern Sulawesi Island, Indonesia.

Materials and Methods

The Notodasus specimens were collected from the southern coasts of China (Fig. 2A) during 2017–2018 and from mix-species seagrass beds of northern Sulawesi Island, Indonesia (Fig. 2B) in May 2014, respectively (for more detail, see Table 1). Indonesian specimens examined in this study were collected with permission of the Ministry of Research and Technology of the Republic of Indonesia (permit no. 135/SIP/FRP/SM/V/2014). In Indonesia, a PVC corer (10 cm in inner diameter) was used to collect sediment samples which were later washed through a 0.5 mm sieve in the field. In China, the Notodasus specimens were collected by means of a grab sampler (surface area 0.05 m2), and then sieved through a 0.5 mm sieve on board. All retained specimens were fixed with 7% diluted formalin in seawater. In the lab, Notodasus specimens were transferred to 70% ethanol.

Table 1 Sampling stations where Notodasus specimens were collected.

Locality	Station	Specimen amount	Date (dd/mm/yyyy)	Longitude	Latitude	Depth (m)	Substrate	
Sulawesi Island (Indonesia)	SGT 1–2	1	23/05/2014	125°06′43″E	1°23′41″N	1	Fine sand	
SGT 3–3	2	25/05/2014	125°06′08″E	1°23′11″N	1	Fine sand	
Guangxi Province (China)	GFC-S11	1	27/10/2017	108°38′15″E	21°37′33″N	12	Mud	
GFC-S24	2	27/10/2017	108°28′29″E	21°32′14″N	11	Mud	
GFC-S31	5	26/01/2018	108°39′52″E	21°41′31″N	8	Mud	
GFC-S33	1	26/01/2018	108°52′41″E	21°34′30″N	7	Muddy sand	
GFC-S17	2	21/04/2018	108°49′42″E	21°31′16″N	7	Mud	
GFC-S18	1	22/04/2018	108°34′41″E	21°35′39″N	9	Muddy sand	

Light microscope images were obtained by means of a Leica M205A stereomicroscope equipped with Leica DFC 550 digital camera. The structure of hooded hooks was observed under a light microscope using oil immersion (Axio Imager Z2; ZEISS, Oberkochen, Germany). A scanning electron microscopy (SEM) analysis was conducted to observe the ultrastructure of abdominal hooks. In brief, the specimens were placed in an ultrasonic chamber with distilled water for 60 s to remove the hoods of the abdominal hooks. The treated specimens were dehydrated and then dried in a drying oven at 60 °C for 5 min. Finally, specimens were mounted on a stub and coated with gold. SEM observations were performed using ZEISS SUPRA 55 SAPPHIRE at Xiamen University, China. The MGSP was used to identify the distribution of glandular areas, following the protocol of Warren, Hutchings & Doyle (1994). Morphological terminology and the characters used for classification follow those of Warren, Hutchings & Doyle (1994).

Type material of several Notodasus species were reviewed from the Natural History Museum of Los Angeles County-Allan Hancock Foundation (LACM-AHF) and the Colección Poliquetologica de la Universidad Autonoma de Nuevo León (UANL). The type material of the two new species described herein are deposited in the Third Institute of Oceanography, Ministry of Natural Resources, Xiamen, China.

Nomenclatural acts

The electronic version of this article in portable document format will represent a published work according to the International Commission on Zoological Nomenclature (ICZN), and hence the new names contained in the electronic version are effectively published under that Code from the electronic edition alone. This published work and the nomenclatural acts it contains have been registered in ZooBank, the online registration system for the ICZN. The ZooBank Life Science Identifiers (LSIDs) can be resolved and the associated information viewed through any standard web browser by appending the LSID to the prefix http://zoobank.org/. The LSID for this publication is: urn:lsid:zoobank.org:pub:6342781B-D33C-4FF8-85BD-37D185FC2403. The online version of this work is archived and available from the following digital repositories: PeerJ, PubMed Central and CLOCKSS.

Results

Systematic account

Class Polychaeta Grube, 1850

Family Capitellidae Grube, 1862

Genus Notodasus Fauchald, 1972

Notodasus Fauchald, 1972: 246–247, Pl.51 fig a–c; Fauchald, 1977: 34; García-Garza, Hernández-Valdez & De León-González, 2009: 810; García-Garza & De León-González, 2011: 35; Magalhães & Bailey-Brock, 2012: 28; García-Garza, De León-González & Harris, 2017: 94, fig. 1; Magalhães & Blake, 2017.

Dodecaseta McCammon & Stull, 1978: 40–43, figs 1–3; Green, 2002: 311.

Type species. Notodasus magnus Fauchald, 1972

Notodasus celebensis sp. nov. Lin, García-Garza & Arbi

urn:lsid:zoobank.org:act:A8E8DAA2-650A-4F10-AD10-3799C447670B

Figs. 3A–3G, 4A–4G and 5G–5H

Figure 3 Photomicrographs of Notodasus celebensis sp. nov.

(A) Thorax and anterior abdomen, lateral view, arrow indicating the separation between thorax and abdomen. (B) Anterior end, lateral view. (C) Prostomium with digitate palpode, lateral view. (D) Longitudinally striated epithelium, lateral view. (E) Transition between thorax and abdomen, lateral view. (F) Anterior abdomen, dorsal view. (G) Posterior part, lateral view. Abbreviations: cc, capillary chaetae; ch, chaetiger; hh, hooded hook; lo, lateral organ; neu, neuropodium; no, notopodium; pal, palpode; per, perstomium. (Photo credit: Junhui Lin).

Figure 4 Holotype of Notodasus celebensis sp. nov. (TIO-BTS-Poly 101).

(A) Anterior 17 chaetigers, lateral view. (B) Anterior end, lateral view. (C) Chaetigers 10–20, dorsal view, showing transition between thorax and abdomen. (D) Chaetigers 10–16, lateral view. (E) Chaetigers 12–18, ventrolateral view. (F) Chaetigers 32–40, lateral view. (G) Neuropodial hook from chaetiger 40. Shading on A–E indicates methyl green staining. Scale bars: A–F, one mm; G, 20 μm. (Drawing by Junhui Lin).

Figure 5 Notodasus species with longitudinally striated epithelium.

Holotype of N. magnus: (A) Anterior end, lateral view. (B) Chaetigers 12–17, dorsal view. Holotype of N. harrisae: (C) Anterior end, lateral view. (D) Chaetigers 11–22, dorsal view. Paratype of N. fauchaldi: (E) Anterior end, dorsal view. (F) Chaetigers 11–17, dorsal view. Holotype of N. celebensis sp. nov., (G) Anterior end, lateral view. (H) Chaetigers 10–19, dorsal view. Methyl green stain: B–H. Scale bars: A–H, one mm. (Photo credit A–F: María E. García-Garza; Photo credit G–H: Junhui Lin).

Etymology. The specific name is derived from the type locality, Sulawesi Island. Celebes is the historical name for modern Sulawesi.

Holotype. TIO-BTS-Poly-101 (sta. SGT3-3), Tanjung Merah Village, the east coast of North Sulawesi (Fig. 2), Indonesia (1°23′41″N, 125°06′43″E), one m depth, fine sand, incomplete, coll. Junhui Lin, May 2014.

Paratype. TIO-BTS-Poly-102 (sta. SGT1-2), one specimen, Kema Village, the east coast of North Sulawesi, Indonesia (1°23′11″N, 125°06′08″E), one m depth, fine sand, incomplete, coll. Junhui Lin, May 2014; TIO-BTS-Poly-103, one specimen, same information as TIO-BTS-Poly-102.

Comparative material examined. N. magnus holotype (LACM-AHF POLY 031), SW Punta Arena, Carmen Island, Gulf of California, from (25°46′00″N, 111°15′00″W) to (25°49′40″N, 111°15′30″W), 29–35 m, March 18, 1949; N. fauchaldi paratype (LACM-AHF POLY 2100) st. E-20 m/BC, Andaman Sea, Thailand (8°30′N, 98°12′E), 21 m, April 22, 1996, muddy sand, coll. SB, ChA; N. harrisae holotype (UANL-6510), Baja California Sur, La Paz Bay, El Tesoro beach (24°15′16.1″N, 110°18′55.4″W), one m, August 1, 2006, coll. MEGG and JALG.

Distribution. Currently known from the Sulawesi Island, Indonesia.

Diagnosis. Prostomium rounded, with digitate palpode. Thorax having one achaetous peristomium and 11 chaetigers with bilimbate capillary chaetae only. First chaetiger biramous. First two abdominal chaetigers with only capillary chaetae in both rami, thereafter, with hooded hooks only. Longitudinally striated epithelium through chaetiger 8. Notopodial lobes completely free along abdomen. Multidentate hooded hooks with four rows of small teeth above main fang. Lateral organs present on thorax and abdomen.

Description

All specimens incomplete. Holotype anterior fragment with 40 chaetigers, 41.0 mm long, 2.8 mm wide in abdomen. Paratypes ranging from 20 mm long by 2.2 mm wide (35 chaetigers) to 25.5 mm long by 2.5 mm wide (39 chaetigers). Body slightly coiled. Color in alcohol yellowish white (Fig. 3A). The dorsal epithelium of anterior abdomen slightly damaged. Prostomium rounded, with digitate palpode (Figs. 3B and 3C). Everted proboscis papillated in paratype (TIO-BTS-Poly-103). Eyespots not observed. Peristomium achaetous, wider than long, same length as first chaetiger, but narrower. Peristomium and first six chaetigers with epithelium longitudinally striated (Figs. 3B–3D, 4A–4B and 5G), chaetigers 7–8 slightly striated, following segments smooth.

Thorax with 11 chaetigers, exclusively with bilimbate capillary chaetae in both rami (Fig. 4A). First chaetiger biramous. All thoracic chaetigers biannulated, 1.5–2 times as wide as long, with deep inter-segmental and clear intra-segmental grooves (Figs. 3B and 4A–4B). Chaetal fascicles inserted just posterior to midline of thoracic segments (Figs. 3B and 3D). Notopodia dorsolateral in anterior thorax, moving dorsally to end of thorax, and neuropodia lateral. Lateral organs present along body, located between noto- and neuropodia; those in thorax closer to notopodia, as small rounded pores; abdominal ones closer to superior neuropodial lobes, as small protuberances (Fig. 3G). Genital pores not observed.

Transition between thorax and abdomen marked by constriction and reduced length of abdominal segments (Figs. 3E–3F, 4C–4D and 5H). First two abdominal segments biannulated, with bilimbate capillary chaetae only and partially developed neuropodial lobes; subsequent abdominal segments with hooded hooks and expanded neuropodial lobes (Figs. 3E–3F, 4D–4E and 5H). Notopodial lobes free along abdomen (Figs. 3F, 4C and 5H), approaching each other on anterior abdomen, but becoming further separated posteriorly. Notopodia with approximately 40 hooded hooks per fascicle. Abdominal neuropodial lobes separated mid-ventrally, extending from ventral area to the dorsolateral region (Figs. 3E and 4C). Neuropodial lobes covered with hooded hooks, leaving enlarged superior neuropodial lobe (Figs. 4D and 4F). Chaetal fascicles in neuropodia with more than 200 hooded hooks. Notopodial and neuropodial abdominal hooded hooks similar along body, with long anterior shaft, angled node, distinct constriction, developed shoulder, and short hood; posterior shaft curved, longer than anterior one, attenuated to terminal end (Fig. 4G). Hooded hooks with four rows of small teeth above main fang (Fig. 4G). Main fang subtriangular, longer than wide.

Branchiae not known, as all examined specimens incomplete. Pygidium not known.

Variations. The holotype is a larger individual than paratypes. Meanwhile, the longitudinally striated epithelium is more evident in the holotype.

Methyl green staining (Figs. 4A–4D and 5G–5H). Thorax uniformly stained light green except by the presence of a medium green transverse band on peristomium. Methyl green stain on first two abdominal chaetigers slightly darker than on thorax. Abdominal chaetigers 3–13 with medium green stain on dorsolateral areas between noto- and neuropodial lobes and ventral areas around neuropodia, a longitudinal mid-ventral band stained with medium green, and light green stain on parapodial tori and lateral organs. Abdominal chaetiger 14 and following ones stained with a light green completely, and mid-ventral band faded.

Notodasus chinensis sp. nov. Lin, García-Garza & Wang

urn:lsid:zoobank.org:act:201C23E8-5FD1-4D12-813D-B0A649A0DC11

Figs. 6A–6F, 7A–7D, 8A–8F and 9A–9B

Figure 6 Photomicrographs of Notodasus chinensis sp. nov.

(A) Thorax and anterior abdomen, lateral view. (B) Anterior thorax, dorsolateral view. (C) Transition between thorax and abdomen, dorsolateral view. (D) Transition between thorax and abdomen, lateral view. (E) Anterior abdomen, showing branchial pores, lateral view. (F) Posterior part of holotype, showing retractile branchiae and lateral organs, lateral view. Abbreviations: bp, branchial pore; br, branchia; cc, capillary chaetae; ch, chaetiger; hh, hooded hook; lo, lateral organ; neu, neuropodium. (Photo credit: Junhui Lin).

Figure 7 Scanning electron micrographs of Notodasus chinensis sp. nov. (TIO-BTS-Poly-106).

(A) Anterior end, dorsolateral view. (B) Transition between thorax and abdomen, dorsolateral view. (C) Capillary chaetae from chaetiger 9. (D) Abdominal hooded hooks. Abbreviations: cc, capillary chaetae; ch, chaetiger; hh, hooded hook; lo, lateral organ. (Photoes by Junhui Lin).

Figure 8 Holotype of Notodasus chinensis sp. nov.

(A) Anterior 17 chaetigers, dorsolateral view. (B) Anterior end, lateral view. (C) Chaetigers 11–18, dorsolateral view, showing transition between thorax and abdomen. (D) Chaetigers 10–20, ventral view. (E) Chaetigers 55–67, lateral view. (F) Neuropodial hook from chaetiger 40. Shading on A–C indicates methyl green staining. Scale bar: A–E = one mm; F = 10 μm. (Drawing by Junhui Lin).

Figure 9 Methyl green staining patterns.

(A–B) Notodasus chinensis sp. nov. (A) Anterior end, dorsolateral view (TIO-BTS-Poly-106). (B) Anterior end, dorsal view (TIO-BTS-Poly-110). (C) N. oraria holotype, anterior end, dorsolateral view. (D–E) N. dexterae holotype. (D) Anterior end, lateral view. (E) Chaetigers 11–17, dorsal view. (Photo credit A–B: Junhui Lin; Photo credit C–E: María E. García-Garza).

Etymology. The specific name is derived from the type locality, Chinese waters.

Holotype. TIO-BTS-Poly-105 (sta. GFC-S31), one specimen, Qinzhou Bay, Guangxi Province (21°41′31″N, 108°39′52″E), eight m, mud, incomplete, coll. Zhong Li, January 26, 2018.

Paratype. 11 specimens: TIO-BTS-Poly-106 (sta. GFC-S31), four incomplete specimens, same information as holotype, one mounted on SEM stub; TIO-BTS-Poly-107 (sta. GFC-S33), one specimen, Qinzhou Bay, Guangxi Province (21°34′30″N, 108°52′41″E), seven m, muddy sand, incomplete, coll. Zhong Li, January 26, 2018; TIO-BTS-Poly-108 (sta. GFC-S11), one specimen, Qinzhou Bay, Guangxi Province (21°37′33″N, 108°38′15″E), 12 m, mud, incomplete, coll. Zhong Li, October 27, 2017; TIO-BTS-Poly-109 (sta. GFC-S24), one specimen, Qinzhou Bay, Guangxi Province (21°32′14″N, 108°28′29″E), 11 m, mud, incomplete, coll. Zhong Li, October 27, 2017; TIO-BTS-Poly-112 (sta. GFC-S17), two specimens, Qinzhou Bay, Guangxi Province, (21°31′16″N, 108°49′42″E), seven m, mud, incomplete, coll. Zhong Li, April 21, 2018; TIO-BTS-Poly-113 (sta. GFC-S18), one specimen, Qinzhou Bay, Guangxi Province (21°35′39″N, 108°34′41″E), nine m, muddy sand, incomplete, coll. Zhong Li, April 22, 2018.

Additional material examined. TIO-BTS-Poly-110, two specimens, Daya Bay, Guangdong Province (22°36′43″N, 114°43′12″E), nine m, mud, incomplete, coll. Junhui Lin, August 30, 2017, one mounted on SEM stub.

Comparative material examined. N. oraria holotype (LACM-AHF POLY 1248), Palos Verdes Peninsula, California, USA, 30–180 m; N. dexterae holotype (LACM-AHF POLY 2190), Naos Island, Panama (8°53′N, 79°33′W), intertidal, sand, incomplete, July 1969.

Distribution. Currently known from shallow subtidal waters of the Qinzhou Bay (Guangxi Province) and Daya Bay (Guangdong Province), the southern coast of China.

Diagnosis. Prostomium conical, with short palpode. Thorax having one achaetous peristomium and 11 chaetigers with bilimbate capillary chaetae only. First chaetiger biramous. First abdominal chaetigers with only capillary chaetae in both rami, thereafter, with hooded hooks only. Tessellated epithelium through chaetiger 5 as well as on dorsum of chaetigers 11 and 12. Notopodial lobes fused dorsally into a raised coalesced lobe on abdominal chaetigers 2–4, while fused but not raised on abdominal chaetigers 5–11. Multidentate hooded hooks with four rows of small teeth above main fang. Lateral organs present on thorax and abdomen. Branchiae present, retractile, arising just above neuropodial hooks. Branchial pores commencing from anterior abdomen.

Description

All specimens incomplete. Holotype anterior fragment with 67 chaetigers, 21.6 mm long, 1.7 mm wide in abdomen (maximum width 1.9 mm at chaetiger 4). Paratypes ranging from 7.7 mm long by 0.6 mm wide in abdomen (23 chaetigers; maximum width 1.0 mm at chaetiger 4) to 87.4 mm long by 2.8 mm wide in abdomen (broken into two parts; more than 100 chaetigers; maximum width 3.7 mm at chaetiger 4). Body slightly coiled. Color in alcohol whitish tan (Fig. 6A). Prostomium conical, with short palpode. Everted proboscis distally ciliated, and proximal portion with numerous minute papillae (Figs. 6A–6B and 8A–8B). Peristomium about same length as first chaetiger, but narrower. Eyespots present, covered by lateral margin of peristomium (TIO-BTS-Poly-106). Peristomium and first five chaetigers with epithelium tessellated, dorsum of chaetigers 11 and 12 slightly tessellated, and remaining segments smooth (Figs. 6A–6B and 8A–8B).

Thorax with 11 chaetigers, exclusively with bilimbate capillary chaetae in both rami (Figs. 6A, 7A–7C and 8A). First chaetiger biramous. Thoracic chaetigers biannulated, being of similar length, 3.5–5 times as wide as long, with clear inter-segmental and intra-segmental grooves (Figs. 6A–6D and 8A–8B). Notopodia dorsolateral in first chaetiger, approaching each other gradually to end of thorax, and neuropodia lateral (Fig. 8A). Chaetal fascicles inserted just posterior to midline of thoracic segments (Figs. 6A–6C, 7A–7B and 8A–8B). Lateral organs evident from posterior thorax, located between noto- and neuropodia; those in posterior thorax closer to notopodia, as small rounded pores (Fig. 6D); those in the abdomen closer to superior neuropodial lobes from chaetiger 12 (first abdominal chaetiger), as small protuberances, protruded above surface in posterior segments (Figs. 6F, 7B and 8E). Genital pores not seen.

Transition between thorax and abdomen marked by constriction and reduced length of first abdominal segment (Figs. 6C–6E, 7B and 8A). First abdominal segments biannulated, with bilimbate capillary chaetae in both rami and partially developed neuropodial lobes; subsequent abdominal segments with hooded hooks and expanded neuropodial lobes (Figs. 6C–6E and 8A). Notopodial lobes fused dorsally into a raised coalesced lobe on abdominal chaetigers 2–4 (Figs. 6C and 8C), while fused but not raised on abdominal chaetigers 5–11. Notopodial fascicles almost touching each other on abdominal chaetigers 2–11, forming a continuous line (Figs. 8A and 8C). From abdominal chaetiger 11, gap between notopodial lobes becoming gradually larger. Neuropodial lobes expanded, separated mid-ventrally (Fig. 8D), extending from ventral area to dorsolateral region (Figs. 6D and 6E). Neuropodial lobes covered with hooks, leaving enlarged superior neuropodial lobe (Fig. 6D). Notopodial fascicles positioned posterior part of segment (Figs. 8A and 8C). Chaetal fascicles with approximately 30 hooks in notopodia and more than 150 hooks in neuropodia. Notopodial and neuropodial abdominal hooded hooks similar along body, with long anterior shaft, developed shoulder, bulbous node, indistinct constriction, and short hood; posterior shaft slightly longer than anterior shaft (Fig. 8F). Four rows of small teeth above main fang (Figs. 7D and 8F). Main fang subtriangular, longer than wide.

Branchiae digitiform in holotype, may be retractile, only observed on some superior neuropodial lobes of abdominal segments (Fig. 6F), arising from a small pore just above neuropodial fascicles. Pygidium not seen.

Variations. All specimens are incomplete, without posterior abdomen. Tessellated epithelium on chaetigers 11 and 12 are more evident in larger specimens. Fused notopodial lobes located on a raised coalesced lobe on abdominal chaetigers 2–4 in larger specimens, while on abdominal chaetigers 2–3 in smaller specimens.

Methyl green staining pattern (Figs. 8A–8C and 9A–9B). Thorax and abdominal segments completely stained light green except that dark green stain from chaetiger 6 to prechaetal area of chaetiger 7, and moderate green stain from chaetiger 11 to prechaetal area of chaetiger 12.

Discussion

On N. celebensis sp. nov.

Among all 10 known Notodasus species worldwide, Notodasus celebensis sp. nov. (Fig. 5G) is most similar to N. magnus (Fig. 5A) and N. harrisae (Fig. 5C) from the Gulf of California, and N. fauchaldi (Fig. 5E) from the Andaman Sea by having longitudinally striated epithelium on thoracic segments, whereas the rest members of the genus bear thoracic segments with tessellated epithelium. However, N. celebensis sp. nov. differs from these three closely related species, based on other morphological characters (Table 2). N. celebensis sp. nov. is distinguished from N. magnus in that: (1) N. celebensis sp. nov. bears rounded prostomium with digitate palpode compared with conical prostomium with short palpode in N. magnus; (2) striated epithelium is present on more thoracic segments in N. magnus than in N. celebensis sp. nov.; (3) abdominal notopodial lobes are completely separated along the abdomen in N. celebensis sp. nov., while in N. magnus, they are fused with a median constriction in anterior abdomen; (4) N. celebensis sp. nov. has abdominal hooks with four rows of small teeth above main fang instead of three rows as in N. magnus. N. celebensis sp. nov. also differs from N. harrisae in that: (1) N. celebensis sp. nov. bears prostomium with digitate palpode and without eyespots compared with prostomium with short palpode and eyespots in N. harrisae; (2) striated epithelium are present on anterior 8 chaetigers of N. celebensis sp. nov. but on the entire thorax of N. harrisae; (3) abdominal notopodial lobes are completely separated along the abdomen in N. celebensis sp. nov. whereas they are fused in the anterior abdomen of N. harrisae; (4) abdominal hooks of N. celebensis sp. nov. have four rows of small teeth above main fang instead of three rows as in N. harrisae. Furthermore, N. celebensis sp. nov. differs from N. fauchaldi in that: (1) N. celebensis sp. nov. bears prostomium with digitate palpode and without eyespots compared with prostomium with short palpode and eyespots in N. fauchaldi; (2) abdominal notopodial lobes are completely separated along the abdomen in N. celebensis sp. nov. whereas they are fused in the anterior abdomen of N. fauchaldi; (3) in anterior abdomen, lateral organs are situated in a pit in N. celebensis sp. nov. but protruded above surface in N. fauchaldi.

Table 2 Comparisons of closely related species in the genus.

Morphological characters	N. celebensis sp. nov.
Holotype	N. fauchaldi
Green (2002)
Paratype	N. harrisae
García-Garza et al. (2009)
Holotype	N. magnus
Fauchald (1972)
Holotype	
Body width in abdomen	2.8 mm	0.7 mm	Two mm	Five mm	
Eyespots	Absent	Present	Present	Absent	
Prostomium	Rounded with palpode	Conical with palpode	Conical with palpode	Conical with palpode	
Thoracic epithelium	Longitudinally striated through chaetiger 8	Longitudinally striated through chaetiger 7	Longitudinally striated through chaetiger 11	Longitudinally striated except for peristomium	
Degree of fused notopodia in anterior abdomen	Completely free	Notopodia fused dorsally but chaetal fascicles separated	Notopodia fused dorsally and chaetal fascicles almost fused	Notopodia fused dorsally with a median constriction, and chaetal fascicles fused	
Dental structure of hooded hooks	Four rows of small teeth above main fang	Four rows of small teeth above main fang	Three rows of small teeth above main fang	Three rows of small teeth above main fang	
Shape of the shaft of hooded hooks	With angled node	With bulbous node	With angled node	With angled node	
Abdominal lateral organs	As a small protuberance in the pits	Protruded above surface	As a small protuberance in the pits	Protruded above surface	
Pygidium	Unknown	Unknown	Unknown	Unknown	
Branchiae	Unknown	Unknown	Evident from chaetiger 60, with around 14 filaments	Evident from chaetiger 61, with around six filaments	
Habitat	One m; fine sand	21–55 m; sandy mud, muddy sand, and sand with shell fragments	0–1 m; fine or coarse sand	29–35 m; mixed sediment of sand, mud, and pebbles	
Type locality	Sulawesi Island, Indonesia	Andaman Sea, Thailand	Gulf of California	Gulf of California	
Reference	This study	García-Garza, De León-González & Harris (2017)	García-Garza, Hernández-Valdez & De León-González (2009)	García-Garza, Hernández-Valdez & De León-González (2009)	

The inhabiting environment is also different: N. celebensis sp. nov. was found in the shallow nearshore seagrass beds characterized by fine sand; N. magnus was collected from soft sediments mixed with sand, mud, and pebbles at depths of 29–35 m; N. harrisae was found to inhabit intertidal and shallow fine sand; and N. fauchaldi was recorded in a variety of sediments at depths of 21–55 m.

As for methyl green staining, the most relevant characteristic of N. celebensis is that it has a dark transverse band on peristomium and medium green stain on dorsolateral areas of abdominal chaetigers 3–13 (Figs. 5G–5H), which is distinct from the other three Notodasus species. Based on the original descriptions of type species, N. magnus has darker prechaetal and postchaetal transverse band on abdominal chaetigers 3–5 (Fig. 5B); N. harrisae has two dark dorsolateral longitudinal bands on abdominal chaetigers 3–22 (Fig. 5D); and N. fauchaldi has dark green stain on dorsum of abdominal chaetigers except for notopodial lobes and lateral organs (Fig. 5F). Moreover, the latter three species have uniform light green stain on anterior thorax, without a dark band on peristomium.

Quite a few monographs and papers dealing with Indonesian polychaetes have been published (Caullery, 1915, 1944; Horst, 1903, 1910, 1912, 1915, 1916a, 1916b, 1917, 1924; Pettibone, 1970, 1971; AI-Hakim & Glasby, 2004; Pamungkas, 2015, 2017). In these publications, seven capitellid genera were taxonomically recorded in Indonesian waters, namely Capitella, Dasybranchus, Mediomastus, Notomastus, Polymastigos, Promastobranchus, and Scyphoproctus. N. celebensis sp. nov., which is newly described from Sulawesi Island, represents the report of Notodasus in Indonesian waters for the first time. The number of capitellid genera in this area rises to eight genera.

On N. chinensis sp. nov.

Notodasus chinensis sp. nov. mostly resembles N. oraria (Fig. 9C) from the waters off California, USA and N. dexterae (Fig. 9D) from the Pacific coast of Panama. These three species share the tessellated epithelium on thoracic segments, the fused notopodial lobes in anterior abdomen, and the mid-ventrally separated neuropodial lobes along abdomen. However, N. celebensis sp. nov. bears tessellated epithelium on the dorsum of chaetigers 11 and 12 and branchial pores commencing from abdominal chaetiger 2, which are not found in other species in the genus. In addition to the above morphological characters exclusive to N. chinensis sp. nov., N. chinensis sp. nov. can be distinguished from N. oraria in that: (1) eyespots are present in N. chinensis sp. nov while absent in N. oraria; (2) thoracic segments have tessellated epithelium on anterior 5 chaetigers in N. chinensis sp. nov. while on chaetigers 1–8 of N. oraria. N. chinensis sp. nov. also differs from N. dexterae in that: (1) lateral organs are situated in a pit in anterior abdomen in N. chinensis sp. nov. while protruded above surface in N. dexterae; (2) abdominal hooks have four rows of small teeth above main fang in N. chinensis sp. nov. instead of five rows of small teeth as in N. dexterae. For more details, see Table 3.

Table 3 Comparisons of closely related species in the genus.

Morphological characters	N. chinensis sp. nov.
Holotype	N. oraria
McCammon & Stull (1978)
Holotype	N. dexterae
García-Garza et al. (2009)
Holotype	
Body width in abdomen	1.7 mm	0.8 mm	One mm	
Eyespots	Present	Absent	Present	
Prostomium	Conical with short palpode	Conical with palpode	Conical with distal palpode	
Thoracic epithelium	Tessellated through chaetiger 5	Tessellated through chaetiger 8	Tessellated through chaetiger 5	
Degree of fused notopodia in anterior abdomen	Notopodial lobes fused and Chaetal fascicles almost touching	Notopodial lobes fused dorsally but chaetal fascicles separated	Notopodial lobes fused dorsally but chaetal fascicles separated	
Dental structure of hooded hooks	Four rows of small teeth above main fang	Four rows of small teeth above main fang	Five rows of small teeth above main fang	
Shape of the shaft of hooded hooks	With bulbous node	With bulbous node	With bulbous node	
Lateral organs	As a small protuberance in the pits but protruded above surface in posterior segments	Protruded	Protruded above surface	
Pygidium	Unknown	Unknown	Unknown	
Branchiae	Present, retractile	Present	Unknown	
Habitat	7–12 m; mud or muddy sand	1–180 m; mud	Intertidal sand	
Type locality	Guangxi Province, China	Waters off California, USA	Naos Island, Pacific coast of Panama	
Reference	This study	García-Garza, De León-González & Harris (2017)	García-Garza, Hernández-Valdez & De León-González (2009)	

In terms of inhabiting environment, N. chinensis sp. nov. is described from shallow subtidal mud or muddy sand (7–12 m), N. dexterae inhabits intertidal sand, and N. oraria is mainly found in muddy sediment at depths of 1–180 m.

Furthermore, N. chinensis sp. nov. has a distinct MGSP: dark green stain on chaetigers 7–8, medium green stain on chaetigers 11–12, and light green stain on the remaining segments (Figs. 9A and 9B). According to the original descriptions of type materials, N. oraria has medium green stain from the postchaetal part of chaetiger 6 to prechaetal part of chaetiger 10, and dark green stain on chaetigers 11 and 12 (Fig. 9C); N. dexterae has medium green stain on chaetigers 9–13, and two dark dorsolateral bands from the third abdominal chaetiger (Fig. 9E).

According to “Checklist of marine biota of China seas” (Liu, 2008), a total of 17 capitellid species was recorded from Chinese waters, represented by 10 genera, including Capitella, Dasybranchus, Heteromastus, Leiochrides, Mastobranchus, Neoheteromastus, Neomediomastus, Notomastus, Parheteromastus, and Rashgua. In this study, Notodasus is recorded for the first time in Chinese waters.

On the generic definition of Notodasus

The genus Notodasus was originally erected by Fauchald (1972), distinct from other capitellid genera mainly by having only capillaries on all 11 thoracic chaetigers as well as on the first two abdominal ones. Since then, eight Notodasus species had been added to the genus, all in agreement with the generic definition. The genus Dodecaseta was initially established by McCammon & Stull (1978), then its definition was expanded by Green (2002) as having first one or two abdominal chaetigers with capillaries instead of first abdominal chaetiger as in the original definition. Of the three known Dodecaseta species, D. eibyejacobseni completely matched the generic diagnosis of Notodasus. The remaining two species, D. oraria and D. fauchaldi, agree well with the generic definition of Notodasus except that they bear abdominal capillaries only on first abdominal chaetiger instead of on first two abdominal chaetigers as in Notodasus. García-Garza, De León-González & Harris (2017) believed that the above morphological difference might be due to the fact that these two Dodecaseta species were described based on immature specimens. It is well known that the replacement of hooded hooks by capillaries occurs in some capitellid genera during ontogeny (Ewing, 1982; Blake, 2000), and the number of chaetigers with capillaries will change until the adult condition is reached. Based on the high morphological similarity, García-Garza, De León-González & Harris (2017) considered Dodecaseta as a junior synonym of Notodasus, without any technical change in the generic definition.

In addition to the presence of capillaries on all 11 thoracic chaetigers and first two abdominal chaetigers, Notodasus bears other distinctive morphological characters: partially developed neuropodia and protruded lateral organs on the last one or two chaetigers with capillaries (Green, 2002). In this study, one of the newly described species, N. celebensis sp. nov., completely matches the generic diagnosis of Notodasus. However, the other species, N. chinensis sp. nov., agrees with the generic diagnosis of Notodasus in most morphological characters except that all examined specimens are characterized by the presence of capillaries on first abdominal chaetiger and absence on the following abdominal segments, irrespective of body size. Given that the specimens of N. chinensis sp. nov. were collected in different seasons (October 2017, January 2018, and April 2018), we believe that N. chinensis sp. nov. has capillaries restricted to chaetigers 1–12, including 11 thoracic chaetigers and first abdominal chaetiger. Therefore, we suggest expanding the generic definition of Notodasus to have capillaries on all 11 thoracic chaetigers as well as on first one or two abdominal chaetigers, to accommodate the new species. To better clarity the chaetal arrangement of the first two abdominal chaetigers during ontogeny, more specimens of Notodasus species at different development stages are required.

Conclusions

Located between the Pacific Ocean and the Indian Ocean, the Central Indo-Pacific region is an important marine biodiversity hotspot with especially rich marine life. In this region, Notodasus is a poorly known group, and prior to this study, there was no taxonomic report of Notodasus species. The description of new Notodasus species from Sulawesi Island and southern China indicates that there is higher diversity within the genus than expected, and this contributes to better understand its diversity worldwide. Besides, this study provides more data about the ecological aspects of Notodasus. However, future efforts should be devoted to the taxonomy of polychaete fauna in this region due to relatively scant information on this group.

Key to Notodasus species (modified from García-Garza, Hernández-Valdez & De León-González, 2009)

1. Epithelium longitudinally striated on all or part of thoracic segments2

– Epithelium tessellated on all or part of thoracic segments5

2. Epithelium longitudinally striated throughout the thoraxN. magnus Fauchald, 1972

– Epithelium longitudinally striated not exceeding chaetiger 93

3. Notopodial lobes completely free throughout the abdomenN. celebensis sp. nov.

– Notopodial lobes fused dorsally in anterior abdomen4

4. Fascicles of notopodial hooded hooks forming a continuous line in anterior abdomen, abdominal neuropodial lobes fused ventrally, hooded hooks with three rows of teeth above main fangN. harrisae García-Garza, Hernández-Valdez & De León-González, 2009

– Fascicles of notopodial hooded hooks clearly separated along abdomen, neuropodial lobes separated mid-ventrally along abdomen, hooded hooks with four rows of teeth above main fangN. fauchaldi (Green, 2002)

5. Epithelium tessellated along the entire thoraxN. hartmanae García-Garza, Hernández-Valdez & De León-González, 2009

– Epithelium tessellated in anterior thorax6

6. Notopodial lobes completely free throughout the abdomen7

– Notopodial lobes fused dorsally in anterior abdomen9

7. Thoracic epithelium tessellated on segments 1–8, hooded hooks with three rows of teeth above main fang and angled node, first two abdominal chaetigers stained dark greenN. salazari García-Garza, Hernández-Valdez & De León-González, 2009

– Thoracic epithelium tessellated on segments 1–6, hooded hooks with two rows of teeth above main fang and indistinct node, first two abdominal chaetigers stained light green or do not stain8

8. Abdominal lateral organs protruded, dark green stain on chaetiger 14 and following segments except for parapodial toriN. eibyejacobseni (Green, 2002)

– Abdominal lateral organs situated in deep pits, no distinct staining patternN. dasybranchoides Magalhães & Bailey-Brock, 2012

9. Posterior neuropodial lobes small, fused mid-ventrally, hooded hooks with four rows of teeth above main fang and angled nodeN. arenicolaHartmann-Schröder, 1992

– Neuropodial lobes separated mid-ventrally10

10. Notopodial fascicles almost touching each other in anterior abdomen; the epithelium tessellated on the dorsum of chaetigers 11–12N. chinensis sp. nov.

– Notopodial fascicles separated in anterior abdomen; chaetigers 11–12 smooth11

11. Fused notopodial lobes without a median constriction, hooded hooks with five rows of teeth above main fangN. dexteraeFauchald, 1973

– Fused notopodial lobes with a median constriction, hooded hooks with four rows of teeth above main fangN. oraria (McCammon & Stull, 1978).

We are grateful to Mr. Heshan Lin and Mr. Zhiyuan Ma for their assistance with field sampling when in Indonesia. We thank Dr. Xikun Song from Xiamen University, China for his assistance in SEM observations, Dr. Kun Liu for the suggestion for preparing Figs. 1 and 2, and Dr. Zhong Pan for editing of the manuscript. We would like to thank Leslie Harris (LACM-AHF) for her help during María Elena García-Garza’s visit to the museum. We also thank Dr. Patricia Gandini, Dr. Wagner Magalhães, and Dr. Camila da Silva for their valuable comments and suggestions on the earlier version of the manuscript.

Additional Information and Declarations

Competing Interests

Author Contributions

Ethics

Data Availability

New Species Registration

The authors declare that they have no competing interests.

Junhui Lin conceived and designed the experiments, performed the experiments, analyzed the data, contributed reagents/materials/analysis tools, prepared figures and/or tables, authored or reviewed drafts of the paper, approved the final draft.

María Elena García-Garza analyzed the data, contributed reagents/materials/analysis tools, prepared figures and/or tables, authored or reviewed drafts of the paper, approved the final draft.

Ucu Yanu Arbi conceived and designed the experiments, performed the experiments, contributed reagents/materials/analysis tools, authored or reviewed drafts of the paper, approved the final draft.

Jianjun Wang conceived and designed the experiments, analyzed the data, authored or reviewed drafts of the paper, approved the final draft.

The following information was supplied relating to ethical approvals (i.e., approving body and any reference numbers):

Indonesian specimens in this study were collected with permission of the Ministry of Research and Technology of the Republic of Indonesia (permit no. 135/SIP/FRP/SM/V/2014).

The following information was supplied regarding data availability:

The raw data is available in Figs. 3–9 and in Tables 2–3. The type specimens examined in this study are deposited in the Third Institute of Oceanography, Ministry of Natural Resources (Xiamen, China), under accession numbers TIO-BTS-Poly 101–103, 105–109, and 112–113.

The following information was supplied regarding the registration of a newly described species:

Publication LSID: urn:lsid:zoobank.org:pub:6342781B-D33C-4FF8-85BD-37D185FC2403,

Notodasus celebensis sp. nov. LSID: urn:lsid:zoobank.org:act:A8E8DAA2-650A-4F10-AD10-3799C447670B,

Notodasus chinensis sp. nov. LSID: urn:lsid:zoobank.org:act:201C23E8-5FD1-4D12-813D-B0A649A0DC11.

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
