# Peer review of "Two new species of Notodasus Fauchald, 1972 (Annelida: Capitellidae) from the Central Indo-Pacific region"

_PeerJ, doi:10.7717/peerj.7638_

## Round 0.1 · original submission · Minor Revisions

Dear authors, both reviewers conclude that your work is an important contribution to the knowledge for the taxonomy of the family Capitellidae and that it has excellent drawings, Congratulations!!!. Although the work is well written, I recommend that it be re-checked by an English speaking colleague to improve the introduction especially. I hope that the authors make the corrections suggested by both reviewers to be able to re-evaluate your paper for publication.

·

Basic reporting

The manuscript described two new species of a small capitellid genus. Both species are from a region in which the genus Notodasus was not previously collected. The species are fully described and the illustrations are of great quality. The authors provide comparisons of their new species with the most morphologically similar congeners and also provide a key to all species described in Notodasus.

Experimental design

No comment

Validity of the findings

No comment

Additional comments

The manuscript is well written and the findings are well justified. I have only provided some minor suggestions to improve the text.
The figures are of great quality but the authors should indicate the position of some features such as prostomium, peristomium, transition between thorax and abdomen and chaetigers 12 and 13. This will facilitate interpretation of the figures.

·

Basic reporting

Even though English is not the mother language of the authors, I think the manuscript is well written; however, I'd suggest you to improve the section "Introduction" to make it a little bit clearer.
I made some correction along this section (pdf file) and a few arrangements on the sentences. Please, check if the meaning of the sentence remained.

Suggestions (also see pdf file):

Line 37: Please, review this number. Currently, they are over 14,000 described species; see new papers (Bleidorn et al., 2015).
Line 49: Improve this sentence, please.
Line 73: I suggest you to improve this sentence. I made some suggestions to make it clearer. Check if the meaning remained the same.
Line 76: Re-write this sentence, please. I didn't understand what you meant. Use comas to make your sentence clearer.
Line 82: The word “yield” means another thing that doesn’t fit into this sentence in my opinion. It’s just a suggestion.
Line 84: I think you should not generalize; there are oceans where this genus has not been studied yet. Still on line 84, the term “discover”, besides highly used on taxonomic papers, it’s not appropriate. We do not "discover" new species, they were there! We have the opportunity to identify them. Again, it’s just my opinion.

Experimental design

The descriptions and discussion of the new species are very good and concise. However, I made some corrections (pdf file) and a few suggests in these sections.

Line 299: The type of epithelium IS a morphological character, so I'd suggest you to include "other morphological cgaracters" on this sentence.
Figure 3: Despite you put the abbreviations on figure 3 I can't see them. Check them, please.
Figure 6: Include abbreviations on legend.
Figure 7: Include abbreviations on legend.
Figure 9: The contrast of these images is not very good; try to edit them. Maybe, do not use a black bottom; try a dark gray if you couldn’t improve the images.

Validity of the findings

This study is very important and helpful for the taxonomy of the family Capitellidae, which has been studying more frequently on the last years, fortunately.

The report of new species is a sign we still have to pay efforts to know and understand our species diversity worldwide. Besides, researches made at poorly studied areas are very important as well.

Additional comments

Very good work with amazing drawings!

---

## Round 0.2 · accepted · Accept

Dear Authors, Thank you for attending to the concerns of the reviewers. I think now your paper is ready for publication. Congratulations!